# Dihydromyricetin Enhances Exercise-Induced GLP-1 Elevation through Stimulating cAMP and Inhibiting DPP-4

**DOI:** 10.3390/nu14214583

**Published:** 2022-11-01

**Authors:** Luting Wu, Min Zhou, Yingquan Xie, Hedong Lang, Tianyou Li, Long Yi, Qianyong Zhang, Mantian Mi

**Affiliations:** Research Center for Nutrition and Food Safety, Chongqing Key Laboratory of Nutrition and Food Safety, Institute of Military Preventive Medicine, Third Military Medical University (Army Medical University), NO. 30th Gao Tan Yan Street, Shapingba District, Chongqing 400038, China

**Keywords:** dihydromyricetin, exercise, GLP-1, gut microbiota, cAMP, DPP-4

## Abstract

The purpose of this study was to examine whether endogenous GLP-1 (glucagon-like peptide-1) could respond to exercise training in mice, as well as whether dihydromyricetin (DHM) supplementation could enhance GLP-1 levels in response to exercise training. After 2 weeks of exercise intervention, we found that GLP-1 levels were significantly elevated. A reshaped gut microbiota was identified following exercise, as evidenced by the increased abundance of *Bifidobacterium*, *Lactococcus*, and *Alistipes* genus, which are involved in the production of short-chain fatty acids (SCFAs). Antibiotic treatment negated exercise-induced GLP-1 secretion, which could be reversed with gut microbiota transplantation. Additionally, the combined intervention (DHM and exercise) was modeled in mice. Surprisingly, the combined intervention resulted in higher GLP-1 levels than the exercise intervention alone. In exercised mice supplemented with DHM, the gut microbiota composition changed as well, while the amount of SCFAs was unchanged in the stools. Additionally, DHM treatment induced intracellular cAMP in vitro and down-regulated the gene and protein expression of dipeptidyl peptidase-4 (DPP-4) both in vivo and in vitro. Collectively, the auxo-action of exercise on GLP-1 secretion is associated with the gut-microbiota-SCFAs axis. Moreover, our findings suggest that DHM interacts synergistically with exercise to enhance GLP-1 levels by stimulating cAMP and inhibiting DPP-4.

## 1. Introduction

It has been estimated that physical inactivity is the fourth leading cause of death in the world, accounting for 6% of all deaths [1]. Exercise is widely regarded as the most cost-effective and efficient method of controlling body weight and preventing the onset, progression, and development of a variety of metabolic diseases. GLP-1 is an incretin, a peptide hormone secreted in the gut by enteroendocrine L cells in response to nutrient intake, which is known to increase insulin secretion, suppress appetite, and slow gastric emptying [2,3]. It has been reported that acute aerobic exercise reduces appetite and reduces post-exercise energy intake [4,5,6]. These changes are regulated, in part, by alterations in appetite-regulating gut hormones such as GLP-1 and ghrelin [7,8]. Several studies have shown that exercise triggered GLP-1 release from L cells, which may be mediated by IL-6 [9,10]. However, it’s not well understood the role and underlying mechanisms of exercise training in GLP-1 secretion.

The gastrointestinal tract (GIT) harbors a quantity of complex and multitudinous microbial communities exceeding 10^14^, known as the gut microbiota [11]. The composition and function of gut microbiota are largely shaped by genetic components and environmental agents, such as diet [12] and prenatal factors [13]. There is mounting evidence from the literature that exercise is a major determinant of alterations in the abundance and diversity of gut microbiota [14]. Athlete microbiomes contained distinct constituent compositions with an increased abundance of *Prevotella*, *Bacteroides*, *Akkermansia*, or *Veillonellaceae* [15,16]. P9 protein, secreted from *Akkermansia muciniphila*, interacts with intercellular adhesion molecule 2 (ICAM-2) to induce GLP-1 secretion [17]. Previous research has shown that elevated abundance of SCFAs-producing bacteria such as *Clostridiales*, *Lachnospira*, and *Roseburia* in the exercise training group [18]. SCFAs stimulate the secretion of GLP-1 from L-cells via binding to G protein-coupled receptors (GPCRs), such as Ffar2 [19].

DHM is a flavonoid found in vine tea that has a variety of medicinal properties. Myricetin (MYR) is the oxidation product of DHM. MYR has been shown to inhibit DPP-4 activity and expression while increasing endogenous circulating GLP-1 [20]. Recent findings indicated that DHM exhibited a better O_2_—radicals scavenging property than MYR [21]. Additionally, numerous flavonoids have also been reported to encourage the release of GLP-1 [22]. Herein, we wondered whether DHM could also raise GLP-1 levels.

Hence, the study intends to investigate the impact of regular exercise on GLP-1 secretion. We hypothesized that exercise would promote GLP-1 secretion via the gut microbiota-SCFAs axis. Experiments with microbiota depletion and fecal microbiota transplantation (FMT) were carried out to further validate the mechanism. Additionally, to determine whether DHM could further enhance the elevation of GLP-1 levels induced by exercise, a mouse model of DHM combined with exercise has been established. We also investigated how DHM affected GLP-1 secretion and degradation in GLUTag cells.

## 2. Materials and Methods

### 2.1. Animal Experimental Design

We obtained male C57BL/6J mice aged 8 weeks from ENSIWEIER Biotechnology Co., Ltd. (Chongqing, China). The mice were housed at the laboratory animal center of Third Military Medical University. All mice were fed with normal chow (Hua Fukang Biological Technology Co., Ltd., Beijing, China) throughout the experiment. All mouse husbandry protocols adhered to the guidelines of the Army Medical University’s Laboratory Animal Welfare and Ethics Committee (Permit Number: AMUWEC20202166).

The following is the description of the experimental design (Figure 1):To investigate the impact of different doses of short-term swimming training on GLP-1 secretion, mice were randomly divided into 4 groups (*n* = 10), sedentary (CON), 30 min swimming training (Ex-30), 60 min swimming training (Ex-60), and 90 min swimming training (Ex-90) groups. The mice were trained to swim for 15 min over the course of 3 days prior to exercise training. Swimming training was held from 8 a.m. to 12 a.m., once a day, and lasted for 2 weeks. All mice were in good health, and no mice died during the experiment period;To investigate the critical role that the gut microbiome played in the positive effects of short-term training on the release of GLP-1, we conducted antibiotic cocktail therapy and FMT experiments:Mice were randomly divided into three groups for the antibiotic cocktail treatment experiments (*n* = 8), sedentary (CON), antibiotic cocktail treatment (ABX), antibiotic cocktail treatment plus 60 min swimming exercise (ABX + EX). A mixture of the four antibiotics-0.5 g/L vancomycin, 1 g/L ampicillin, 1 g/L metronidazole, and 1 g/L neomycin sulfate-was added to drinking water. Antibiotic treatment and swimming training were carried out simultaneously for 2 weeks. Swimming training was carried out according to experiment 1;For FMT experiments, fecal samples were collected from CON and Ex-60 group mice of experiment 1 at the end of the intervention. Fecal (20 mg) was dissolved in 1 mL of saline, vortexed to mix for 3 min, and then centrifuged at 4 °C for 3 min to separate the microbial supernatant. After 2 weeks of antibiotic cocktail treatment, microbiota-eliminated mice were given 200 μL of the microbial supernatant by oral gavage for another 2 weeks, donor (CON) and donor (Ex-60) group (*n* = 8);To investigate the effects of combined intervention (DHM and exercise), 3 groups of mice (*n* = 8) were randomly assigned, sedentary (CON), 60 min swimming training (EXE), 60 min swimming training plus DHM treatment (EXE + DHM). Swimming training was carried out according to experiment 1. DHM was orally given once a day for four weeks at a dose of 100 mg/kg body weight per feeding. The EXE + DHM mice received DHM intragastric administration for 2 weeks, and CON and EXE mice received the same volume of solvent (saline). Then, EXE and EXE + DHM mice underwent swimming training for another 2 weeks, while the gavage intervention remained the same as the first 2 weeks. DHM (purity  ≥  98%) was purchased from Must Bio-Technology Co., Ltd. (Chengdu, China).

Body weight, water intake, and food intake were all tracked on a regular basis. Fecal samples were collected at the start and end of the experiment. At the end of the experiment, mice were euthanized and dissected to collect serum, colon, and small intestine. All samples were kept separately at −80 °C until subsequent analyses.

### 2.2. Determination of GLP-1 Concentrations

GLP-1 concentrations in serum, cell culture supernatants, and cell lysates were analyzed using a mouse GLP-1 ELISA Kit (Jianglai Biological, Shanghai, China). The manufacturer’s instructions were followed in regard to every step.

### 2.3. Histological Analysis

Immunohistochemistry After being dehydrated and embedded in paraffin, fresh colon samples were fixed in 4% paraformaldehyde. Immunohistochemistry was done following the manufacturer’s protocol. Approximately 5 μm-thick slices were cut and incubated with antibodies against GLP-1 (Servicebio, 1:500) at room temperature for two hours. The secondary antibody used was enhanced goat anti-mouse IgG polymer. The DAB chromogen substrate was added next. The slices were counterstained with Haematoxylin and allowed to air dry before visualizing;Immunofluorescence Tissue sections of colon samples were prepared following the immunohistochemistry procedure. Approximately 5 μm-thick slices were cut and incubated with antibodies against GLP-1 (Servicebio, 1:500) overnight at 4 °C and DAPI for 30 min at room temperature. The slices were then incubated with appropriate conjugated secondary antibodies for 20 min. After the removal of unbound secondary antibodies by washing them in phosphate-buffered saline, imaging was performed.

### 2.4. Fecal Specimen DNA Extraction

Total fecal specimen DNA was extracted from each fecal specimen using a TIANamp Stool DNA Kit (Tiangen Biotech (Beijing) Co., Ltd., Beijing, China). The manufacturer’s instructions were followed in regard to every step. Thermo Scientific NanoDrop 2000 spectrophotometer was used to determine DNA concentration and purity.

### 2.5. Microbiome 16S rRNA Sequencing

As previously mentioned, the extraction and concentration of DNA from fecal specimens were determined. The quality of the DNA was determined using 1% agarose gel electrophoresis. With primer pairs 338F (5′-ACTCCTACGGGAGGCAGCAG-3′) and 806R (5′-GGACTACHVGGGTWTCTAAT-3′), the V3-V4 hypervariable region (from the forward 338F to the reverse 806R) of the bacterial 16S rRNA gene was amplified by an ABI GeneAmp^®^ 9700 PCR thermocycler (ABI, Waltham, MA, USA). The sequencing work was performed by Majorbio Bio-Pharm Technology Co. Ltd. (Shanghai, China). The Appendix A contain detailed instructions for processing raw sequencing data.

The correlation among SCFAs and microbial populations was conducted by Pearson correlation analysis using R software version 3.5.1 and RStudio (R Package), and a Pearson correlation of 0.5 or greater and a *p*-value < 0.05 was considered as a significant correlation.

### 2.6. Measurement of SCFAs by GC-MS

Fecal samples (about 100 mg) were ground with 1 mL of distilled water with phosphoric acid (0.5%) and internal standards (50 μg/mL, 2-ethylbutanoic acid) using a freezing grinder at 50 Hz for 3 min and sonicated in an ice bath for 30 min. The mixture was left standing at 4 °C for 30 min and centrifuged for 15 min at 13,000× *g* (4 °C). The supernatant was collected and then extracted by Majorbio Bio-Pharm Technology Co., Ltd. (Shanghai, China). The extraction solutions were detected using an 8890B-5977B GC/MSD (Agilent Technologies Inc., Santa Clara, CA, USA). The chromatographic separation was carried out on an HP-FFAP capillary column (30 m × 0.25 mm × 0.25 μm, Agilent J&W Scientific, Folsom, CA, USA). The GC conditions were as follows: carrier gas (helium, 99.999%, flow rate 1 mL/min), injection temperature 260 °C, injection volume 1 µL in split mode with split ratio 10:1. The column temperature program was: initial temperature (80 °C), raised to 120 °C at a rate of 40 °C/min, increased to 230 °C at a rate of 10 °C/min, and held at this temperature for 3 min. The MS parameters were as follows: an ion source, electron impact (EI) at 230 °C; quadrupole temperature at 150 °C; transfer line temperature at 230 °C ionization energy, 70 eV; ion source temperature, 230 °C; and selected ion monitoring (SIM) mode. All SCFAs standards were purchased from Sigma-Aldrich (St. Louis, MO, USA) or Shanghai Wokai Biotechnology Co., Ltd. (Shanghai, China).

### 2.7. Cell Culture

GLUTag cell, a murine enteroendocrine cell line, was obtained from BLUEFBIO (Shanghai, China). The Dulbecco’s modified Eagle medium (DMEM, Biological Industries, Israel) was used to cultivate the Murine GLUTag cells contained 25 mmol/L glucose, 10% (*v*/*v*) fetal bovine serum (Biological Industries) and 1% (*v*/*v*) penicillin-streptomycin solution (Beyotime) (37 °C, 5% CO_2_). At 80–90% confluence, cells were routinely passaged using trypsin EDTA solution A (Biological Industries).

### 2.8. Assessment of Cell Viability by Cell Counting Kit-8 (CCK8)

CCK8 (Beyotime, Shanghai, China) assay was performed to determine the cell viability of GLUTag cells. In 96-well culture plates, cells were seeded with culture medium and incubated for one night. After being treated with DHM (0, 2.5, 5, 10, 20, 40, 80 μM) for 2 h, cells were incubated for 2 h at 37 °C with an additional 10 μL of CCK8. Finally, a microplate reader was used to measure the absorbance at a wavelength of 450 nm.

### 2.9. Measurement of DPP-4 Content and Activity

DPP-4 content in colon and cell lysates was analyzed using a DPP-4 ELISA assay kit (Jianglai Biological, Shanghai, China). DPP-4 content was expressed as pg/mL. DPP-4 activity in serum and cell medium supernatants was analyzed using a DPP-4 activity assay kit (Sigma-Aldrich, St. Louis, MO, USA). DPP4 activity was reported as pm/min/mL = microunit/mL, one unit of DPP4 was the amount of enzyme that will hydrolyze the DPP4 substrate to yield 1.0 μm of the fluorescent product (7-amino-4-methyl coumarin) per minute at 37 °C. The manufacturer’s instructions were followed in regard to every step.

### 2.10. Measurement of Second Messengers (Ca^2+^, cAMP, IP_3_)

Intracellular Ca^2+^ contents were detected with Fluo-4 AM (Beyotime, Shanghai, China) in GLUTag cells. cAMP concentrations were analyzed using a cAMP-Glo™ assay kit (Promega, Beijing, China) in lysates of GLUTag cells. Intracellular IP_3_ content was measured using an ELISA kit (Elabscience, Wuhan, China) in lysates of GLUTag cells. The manufacturer’s instructions were followed in regard to every step.

### 2.11. Real-Time PCR

Total RNA was extracted from colon or GLUTag cells using the Trizol reagent (Invitrogen, Shanghai, China) according to the manufacturer’s directions. Reverse transcription of mRNA was performed using PrimeScript RT Master Mix (Takara, Beijing, China) according to the manufacturer’s directions. Gene-specific primers were designed by Sangon Biotech Co., Ltd. (Shanghai, China). Real-time PCR was conducted with TB Green Premix Ex Taq II (Takara, Beijing, China) according to the manufacturer’s instructions. The mRNA expression levels were normalized to those of Actb, which was used as an internal control. Gene-specific primer sequences are listed in Table 1.

### 2.12. Data Processing

The statistical evaluations were performed using GraphPad Prism Version 8.3.0. The mean ± standard error of the mean (SEM) was used to represent data, with a statistically significant difference determined as *p* < 0.05. The unpaired *t*-test, 1-way analysis of variance (ANOVA) followed by Tukey’s multiple comparisons test, and 2-way ANOVA tests followed by Sidak’s multiple comparisons test were used for statistical analysis.

## 3. Results

### 3.1. Effects of Swimming Training on GLP-1 Secretion

To ascertain whether exercise training stimulated GLP-1 secretion, different doses of swimming training models were established. As shown in Appendix A, the final body weights of swimming training mice were remarkedly lower than that of control mice. The body weight gain of swimming training mice was also significantly decreased compared to CON mice (Appendix A). However, there were no differences in 24 h food intakes and water intakes (Appendix A). Compared with the control group, swimming training significantly reduced the weight of inguinal white adipose tissue (iWAT) (Appendix A). The soleus muscle mass was increased in the Ex-90 group (Appendix A), while two-week swimming training had no discernible effects on gastrocnemius and tibialis anterior muscle mass (Appendix A). At the end of the training, GLP-1 concentrations were significantly increased in the Ex-60 and Ex-90 groups (Figure 2A). Gcg, encoding GLP-1, was also upregulated by swimming training (Figure 2B). Western blotting analysis of colon samples revealed that the expression of GLP-1 protein notably increased in the Ex-60 and Ex-90 groups (Figure 2C,D). What’s more, the results of immunohistochemistry staining showed that the proportion of GLP-1 positive area was higher in the Ex-60 group than in the CON group (Figure 2E,F). Collectively, these results confirm that two-week swimming training of 60 min increased GLP-1 content both in the colon and serum.

### 3.2. Effects of Swimming Training on Gut Microbiota

To probe whether swimming training could alter the composition of the gut microbial community, fecal samples were collected at week 0 (_w0) and week 2 (_w2) in the control and Ex-60 groups. In order to facilitate analysis in the cloud platform system, we marked the control group as CON and the Ex-60 group as EXE. To gauge community diversity and richness, the OTU-level alpha diversity indices Sobs and Shannon index was calculated. As shown in Figure 3A,B, EXE_w2 mice had higher Sobs and Shannon index than CON_w2 mice. The similarity or differences in sample community composition were analyzed using principal coordinates (PCoA) analysis on the OUT level. As shown in Figure 3C, the samples from the EXE_w2 group showed a significantly distinct change compared with the other three groups, which indicated swimming training induced significant variability in gut microbiota profiles. Notably, the Wilcoxon rank-sum test at the phylum level revealed reduced proportions of *Firmicutes* and *Campilobaterota*, increased proportions of *Actinobacteriota*, *Cyanobacteria,* and *Deferribacterota* in the EXE_w2 mice, also indicated a decrease in the ratio of *Firmicutes/Bacteroidetes* (Figure 3D–F and Appendix A). At the genus level, the relative abundance of *Bifidobacterium*, *Alistipes*, *Odoribacter*, *Rikenellaceae_RC9_gut_group*, *Enterorhabdus*, *Muribaculum*, and *Lactococcus* was increased in the EXE_w2 group compared with the CON_w2 group (Figure 3F,H and Appendix A). At the genus level, the relative abundance of *Bacteroides*, *Helicobacter*, *Alloprevotella,* and *Lachnoclostridium* was decreased in the EXE_w2 group compared with the CON_w2 group (Appendix A).

### 3.3. Effects of Swimming Training on Short-Chain Fatty Acid Profiles in Fecal

Since swimming training altered the composition of gut microbiota, the next step in our investigation was to see whether SCFA concentrations were changed in feces. The concentrations of eight SCFAs (acetic acid, propanoic acid, butanoic acid, isobutyric acid, valeric acid, isovaleric acid, hexanoic acid, and isohexanoic acid) were determined using GC-MS. Swimming training notably increased propanoic acid and butanoic acid levels, while the levels of other SCFAs displayed no differences between the two groups (Figure 4A). To further screen out the gut microbiota associated with propionic acid and butanoic acid levels, Pearson correlation analysis was performed. The results of the correlation heatmap analysis showed a significant correlation between the level of butanoic acid and *Actinobacteriota*, *Patescibacteria*, and *Firmicutes* (Figure 4B). Furthermore, the level of propionic acid was significantly correlated with *Bacteroidota*, *Patescibacteria*, *Firmicutes*, and *Desulfobacterota*. At the genus level, significant positive correlations were observed between the level of butanoic acid and *Alistipes*, *Enterorhabdus*, *Rhodococcus*, and *Ruminiclostridium*, while significant positive correlations were observed between the level of propionic acid and Pseudomonas, *Burkholderia-Caballeronia-Paraburkholderia* (Appendix A). On the one hand, it has been well documented that SCFAs could serve as GLP-1 secretagogues by binding to GPCRs, such as Ffar2 (23). On the other hand, SCFAs selectively promote differentiation and consequently increase the number of L-cells (24). Hence, we measured the expression of L cell proliferation markers in colon tissues. Our results showed swimming training significantly induced the gene expression of ND1, Arx, and Ngn3 (Figure 4C).

### 3.4. Gut Microbiota Is Essential for Exercise-Induced GLP-1 Secretion

In order to determine the role of gut microbiota for exercise-induced GLP-1 secretion, we carried out microbiota depletion and FMT experiments. Two-week of antibiotic treatment significantly reduced fecal bacterial DNA concentration (Figure 5A). There was no difference in serum GLP-1 levels between the ABX group and ABX + EX group (Figure 5B). Meanwhile, the protein expression of GLP-1 was similar in the colon between the two groups (Figure 5C). We further measured the gene expression of Gcg and L cell proliferation markers (ND1, Arx, Foxa1, Foxa2, Ngn3). As shown in Figure 5D,E, after treatment with antibiotics, the effects of exercise on promoting the expression of Gcg and L cell proliferation markers disappeared. In addition, to further verify that exercise worked through the gut microbiota, we transplanted the fecal gut microbiota from the Ex-60 mice into the antibiotic-treated, microbiota-eliminated mice. The results indicated that the effects of exercise on the GLP-1 secretion were transferable by the transplantation of the gut microbiota (Figure 5F–I). The recipients of microbiota from the Ex-60 mice exhibited a higher expression of GLP-1 compared to that of microbiota from the CON mice (Figure 5F–H). To summarize, these data suggest that the auxo-action of exercise on GLP-1 secretion was largely dependent on the gut microbiota.

### 3.5. Dihydromyricetin Changes the Composition of Gut Microbiota in Exercised Mice

To investigate whether DHM could further modulate the gut microbiota composition, we performed 16S rRNA sequencing on the feces of EXE and EXE + DHM mice. The Sobs index, indicating community richness, displayed no obvious difference between the two groups (Figure 6A). The Shannon index, indicating community diversity, was significantly increased in EXE + DHM group (Figure 6B). Besides, the results of PCoA analysis showed no significant difference in β diversity of gut microbiota between the two groups (Figure 6C). Next, we explored the microbial species that exhibited differences in community abundance between the two groups (Figure 6D,E). At the species level, the relative abundance of *uncultured_Bacteroidales_bacterium_g__Alloprevotella*, *uncultured_bacterium_g__Muribaculum*, *uncultured_Bacteroidales_bacterium_g__Parabacteroides*, *Rikenella_microfusus_DSM_15922*, *Helicobacter_hepaticus*, and *Lactobacillus_murinus* was higher in the EXE + DHM group compared with the EXE group (Figure 6F,H). At the species level, the relative abundance of *ncultured_bacterium_g__norank_f__norank_o__Gastranaerophilales*, *unclassified_g__Ruminococcus*, *uncultured_bacterium_g__Candidatus_Saccharimonas*, *unclassified_g__Marvinbryantia*, and *unclassified_g__NK4A214_group* was significantly decreased in the EXE + DHM group compared with the EXE group (Figure 6F,H). At the genus level, the relative abundance of *Alloprevotella*, *Muribaculum*, *Parabacteroides*, *Rikenella*, and *Candidatus_Stoquefichus* was higher in the EXE + DHM group compared with the EXE group (Figure 6F,G). While, the relative abundance of *norank_f__norank_o__Gastranaerophilales*, *Ruminococcus*, *Candidatus_Saccharimonas*, and *NK4A214_group* was significantly decreased in the EXE + DHM group compared with the EXE group (Figure 6F,G). 

### 3.6. Dihydromyricetin Interacts Synergistically with Exercise Intervention to Enhance GLP-1 Levels

DHM treatment has previously been shown in our lab to reduce insulin resistance and improve glucose tolerance in obese mice [23]. Hence, we conjectured whether DHM could promote the release of GLP-1. Our results indicated that DHM combined with exercise exhibited higher GLP-1 levels both in serum and colon samples, compared with the exercise alone (Figure 7A,C,D). However, it was puzzling that Gcg mRNA did not show differential expression between EXE and EXE + DHM groups (Figure 7B). Furthermore, our findings indicated that DHM intervention had no impact on the quantity of SCFAs in the stools of trained mice (Figure 7E). Likewise, gene expression of the L cell proliferation markers (ND1, Arx, Foxa1, Foxa2, Ngn3) revealed no significant difference between EXE and EXE + DHM mice (Figure 7F). Taking the above results together, DHM elevated GLP-1 levels, possibly independent of promoting GLP-1 production.

### 3.7. Dihydromyricetin Stimulates GLP-1 Release and Reduces GLP-1 Degradation In Vivo and In Vitro

Next, we speculated whether DHM increased GLP-1 levels by stimulating GLP-1 release and/or reducing GLP-1 degradation because GLP-1 is a peptide hormone with a short half-life and is easily degraded. DPP-4 is the main cause of GLP-1 degradation, which exists in two forms, a circulating protein and a membrane-spanning cell surface protein [24]. To test our hypothesis, we first examined the content and activity of DPP-4. Exercise alone did not alter the colonic expression of the DPP-4 mRNA and protein compared to the CON group, while DHM combined with exercise remarkedly decreased the gene and protein expression of DPP-4 compared to the exercise alone (Figure 8A,B). Moreover, DPP-4 activity in the serum of the EXE + DHM mice was significantly lower than that of EXE mice (Figure 8C). Next, we explored whether DHM could stimulate GLP-1 secretion in GLUTag cells. Treatment of GLUTag cells with DHM treatment did not affect cell viability (Figure 8D). GLP-1 secretion was modestly increased 2 h after direct DHM (10 μM) stimulation in GLUTag cells (Figure 8E). Then, 10 μM of DHM was chosen to perform the subsequent assays. There was an elevation in intracellular GLP-1 content after DHM intervention (Figure 8F). The gene expression of DPP-4 was also measured in GLUTag cells. DHM was shown to downregulate DPP-4 mRNA expression (Figure 8G). What’s more, there was a significant reduction in intracellular DPP-4 content and DPP-4 activity with DHM treatment (Figure 8H,I). Next, we examined the effect of DHM treatment on intracellular second messengers. We found that DHM significantly increased cAMP concentrations without altering Ca^2+^ and IP_3_ levels in GLUTag cells (Figure 8J–L). In conjunction with the above findings, DHM may stimulate GLP-1 release by increasing cAMP content and decreasing GLP-1 degradation by inhibiting DPP-4 expression, thereby raising GLP-1 levels.

## 4. Discussion

Scientific studies have confirmed that exercise has a variety of health benefits, including physiological, cardiovascular, and mental health benefits. Numerous studies have shown that single bouts of exercise reduced orexigenic hormone acylated ghrelin, increased anorexigenic signals GLP-1 and PYY, and subsequently regulated appetite [7,8]. Moreover, chronic exercise has been reported to elevate GLP-1 secretion [25]. These are critical reasons for exercising to reduce fat mass and control body weight. Although decades of research have been done, the underlying mechanisms through which exercise training induces GLP-1 secretion are still unknown. In this study, we probed the effect of varying doses of swimming training on GLP-1 secretion in vivo, as well as the underlying mechanisms involving the gut microbiota and SCFAs. Furthermore, DHM treatment increased exercise-induced GLP-1 levels in both the serum and the colon. To the best of our knowledge, this research is the first to demonstrate that the gut microbiota may mediate the critical role of exercise in GLP-1 release.

Several studies have reported that acute exercise markedly increased GLP-1 secretion [26,27,28,29]. IL-6, a kind of myokine, is released from skeletal muscle in response to exercise [30]. Helga Ellingsgaard et al. showed that skeletal-muscle-derived IL-6 mediated an increase in GLP-1 secretion induced by exercise [9]. However, it is uncertain whether chronic exercise could induce GLP-1 secretion. Here, our study showed that two-week swimming training notably elevated GLP-1 levels in serum and colon. Gcg, a gene encoding GLP-1, was also upregulated by swimming training. Collectively, these results indicate swimming training may induce GLP-1 release. So, what’s the mechanism?

As a sensor of environmental stimulation and changes, gut microbiota alters accordingly, such as gut flora composition and gut flora-associated metabolites, which can directly act on intestinal epithelial cells and distal tissue cells, leading to functional changes in gene expression and metabolism, and thus, phenotypic alterations. It has been demonstrated that exercise, independent of diet or other factors, could alter the composition and diversity of gut microbiota [31]. Consistent with our results, swimming training improved bacterial richness (Sobs index) and diversity (Shannon index). Furthermore, exercise decreased the ratio of Firmicutes/Bacteroidetes, which is regarded as a sign of obesity [32]. At the same time, our study showed that the genus *Akkermansia* was not responsive to swimming training. We demonstrated that in EXE_w2 mice, at the genus level, the changes are distinguished by a significant increase in *Bifidobacterium*, *Alistipes*, *Odoribacter*, *Rikenellaceae_RC9_gut_group*, *Enterorhabdus*, *Muribaculum*, *Lactococcus* and a reduction in *Helicobacter*, *Alloprevotella, Lachnoclostridium*. Among them, *Bifidobacterium* and *Lactococcus* are widely used in probiotic supplements, promoting the digestibility of nutrients, such as oligosaccharides and indigestible dietary fiber [33]. SCFAs are the final product of bacterial fermentation of carbohydrates and dietary fibers [34]. What’s more, *Alistipes*, expressed cobalamin-binding methylmalonyl-CoA mutase and/or methylmalonyl-CoA epimerase, plays an important role in SCFA production [35]. Hence, we conducted GC-MS to address the changes in the SCFA content of the fecal material. Coordinated with alterations in gut microbiota, exercise increased the content of propanoic acid and butanoic acid. Intestinal L cells exhibit high expression of several G-protein-coupled receptors (GPCRs), such as Ffar1, Ffar2, GPR119, and GPBAR [36]. On the one hand, previous studies have indicated that SCFAs triggered the secretion of GLP-1 in enteroendocrine L cells through Ffar2, which is predominantly expressed in colonic L cells [19,37]. On the other hand, SCFAs have also been proven to increase L cell numbers [38]. We determined the expression of L cell proliferation markers (ND1, Arx, Foxa1, Foxa2, Ngn3) in the colon. We found that the gene expression of ND1, Arx, and Ngn3 was upregulated by swimming. Activated FFar2 initiates signaling through the Gαq pathway to elevate cytosolic Ca^2+^, which stimulates the exocytosis of GLP-1 from intestinal L cells [39,40]. Taken together, our data indicated that swimming training might stimulate GLP-1 secretion through the gut microbiota-SCFAs-Ffar2 axis. In order to elucidate gut microbiota is essential for exercise-induced GLP-1 secretion, we performed antibiotic-induced microbiota depletion and FMT experiments. Exercise-induced GLP-1 secretion was significantly negated by treatment with antibiotics, which was again reversed by the transplantation of gut microbiota. Collectively, these findings suggested that the effects of exercise on GLP-1 release were primarily determined by the gut microbiota.

Due to hectic work schedules and other factors, modern people rarely participate in and adhere to physical exercise. In addition, GLP-1 has a short biological half-life, which is rapidly degraded by DPP-4 [41]. Once degraded by DPP-4, GLP-1 loses its bioactivity rapidly, in turn, can’t exert a regulatory role in the distal tissues and cells. Therefore, the development of DPP-4-resistant peptide analogs has been required to generate robust GLP-1 efficacy. Previous research revealed that DHM had poor absorption into the blood and was unstable in the intestinal environment, implying that DHM is metabolized and eliminated in the intestinal tract [42]. Recent research has shown that DHM regulates fecal bile acid metabolism by targeting the intestinal flora [43,44]. DHM treatment remarkedly increased the level of probiotics, such as *Bifidobacterium*, *Lactobacillus*, and *Akkermansia* [43,44]. What’s more, DHM treatment decreased the ratio of Firmicutes/Bacteroidetes, accompanied by a relative decrease in the *Firmicutes* phylum and a relative increase in the *Bacteroidetes* phylum [45]. Thus, it’s quite possible that DHM exerts a crucial role in the intestinal tract. Our results indicated that DHM combined with exercise exhibited higher GLP-1 levels both in the colon and circulation compared with the exercise alone. Likewise, we also observed that DHM stimulated GLP-1 secretion from GLUTag cells. Although DHM intervention further altered the gut microbiota composition in exercised mice, showing increased *Alloprevotella* and *Rikenella* genus, DHM did not demonstrate an impact on the exercise-induced elevation of *Bifidobacterium*, *Lactococcus*, and *Alistipes* genus. We also examined the SCFA content of feces. DHM did not increase the amount of eight SCFAs in exercised mice. We, therefore, speculated that there might be other mechanisms by which DHM elevated GLP-1 levels. Recently, it has been reported that MYR could inhibit the expression and activity of DPP-4 from enhancing circulating GLP-1 [20]. DHM and MYR exhibit several similar bioactive properties. At the same time, DHM has been shown to have superior pharmacological properties to MYR [46,47]. Herein, on the one hand, we speculated whether DHM could also inhibit the activity and/or expression of DPP-4. In our study, DHM treatment decreased the expression of DPP-4 in the colon and serum, as well as DPP-4 activity in serum. Additionally, we also observed that DHM restrained DPP-4 expression in vitro. On the other hand, activation of GPCRs can stimulate GLP-1 secretion from L cells [48]. The change of intracellular second messengers Ca^2+^, cAMP, or IP_3_ can reflect the activation of GPCRs [48]. DHM significantly induced intracellular cAMP without changing Ca^2+^ and IP_3_. Previous research has demonstrated that cAMP stimulated GLP-1 secretion via the downstream signaling molecule PKA-CREB [49]. When combined, DHM may reduce the first step of GLP-1 degradation by inhibiting DPP-4 expression in the colon, stimulating GLP-1 secretion by cAMP signaling, thereby increasing the GLP-1 levels in circulation, and the underlying mechanisms warrant further investigation.

## 5. Conclusions

In conclusion, the present research demonstrates that the auxo-action of exercise on GLP-1 secretion is associated with an increase in propanoic acid and butanoic acid due to an increase in the abundance of SCFAs-producing gut microbiota in mice. Additionally, our findings indicate that DHM interacts synergistically with exercise intervention to enhance GLP-1 levels through stimulating cAMP and inhibiting enteric DPP-4. Our research extends our understanding of pharmacological efficacies for DHM and provides evidence for the development of DHM as an exercise enhancer.

## Figures and Tables

**Figure 1 nutrients-14-04583-f001:**
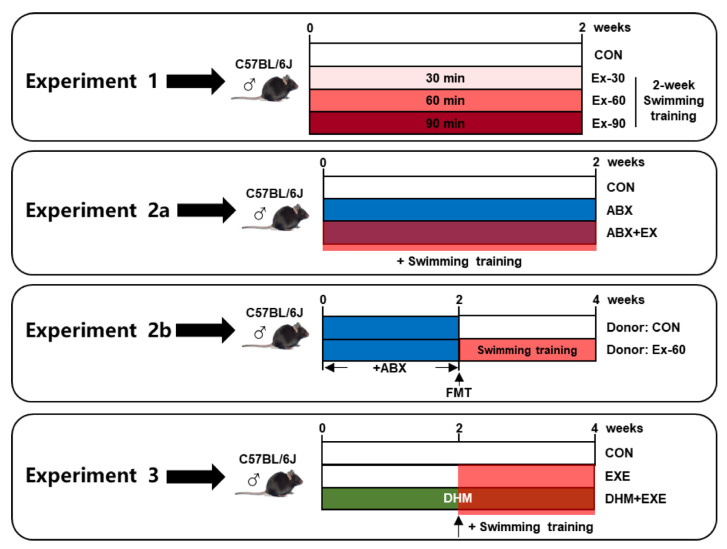
Schema showing the experiment design. Light red indicates 30 min swimming training, red indicates 60 min swimming training, dark red indicates 90 min swimming training, blue indicates ABX treatment, and green indicates DHM treatment.

**Figure 2 nutrients-14-04583-f002:**
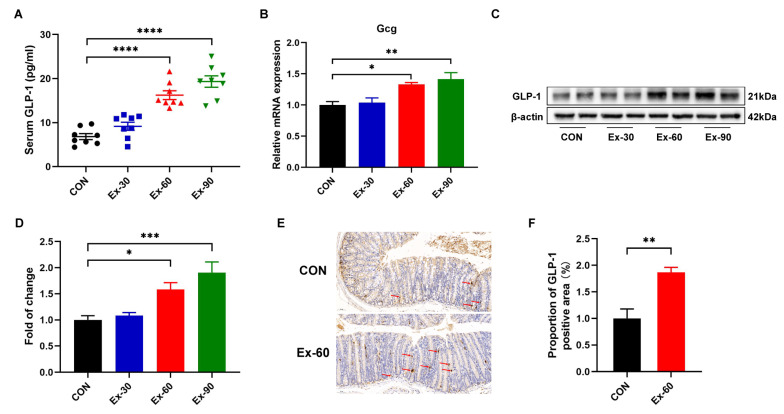
Two-week swimming training promotes GLP-1 secretion. (**A**) GLP-1 levels were measured using an ELISA kit in serum (*n* = 8–10). (**B**) qPCR analysis of the expression of the Gcg gene in colon samples. (**C**,**D**) Representative western blot of GLP-1 from colon samples and quantification. (**E**,**F**) Immunohistochemistry staining of GLP-1 for colon samples and analysis of the proportion of GLP-1 positive area using Image J. Scale bars, 50 μm. Data are expressed as the mean ± SEM. One-way ANOVA: (**A**,**B**,**D**). Unpaired *t*-test: F. * *p* < 0.05, ** *p* < 0.01, *** *p* < 0.001, and **** *p* < 0.0001.

**Figure 3 nutrients-14-04583-f003:**
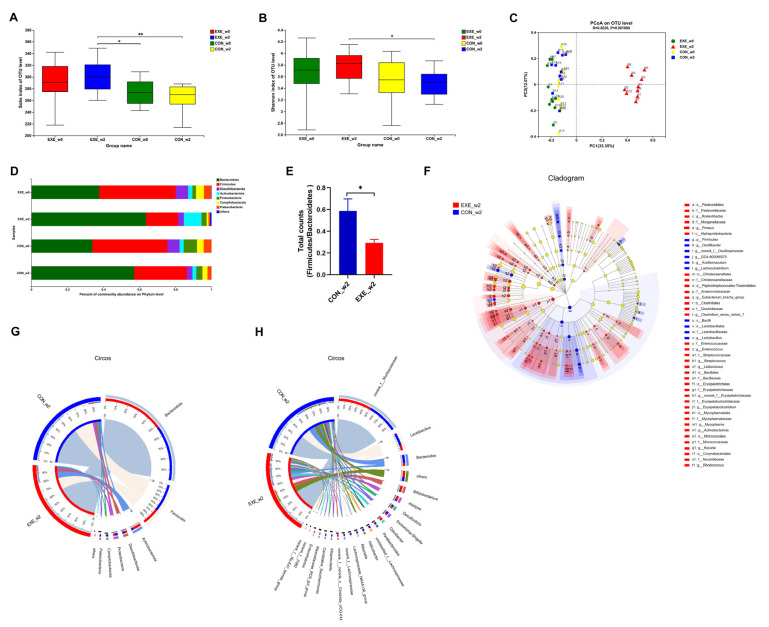
Two-week swimming training alters gut microbiota profile. (**A**,**B**) The OTU-level alpha diversity indices such as Sobs (**A**) and Shannon (**B**) index. (**C**) Weighted UniFrac PCoA analysis on OUT level. (**D**) Community composition analysis on phylum level. (**E**) The ratio of Firmicutes/Bacteroidetes (total counts of members from the phylum level) was calculated. (**F**) Linear discriminant analysis effect size (LEfSe) analysis from phylum to genus level. (**G**,**H**) The relationships between samples and bacteria at the phylum (**G**) and genus (**H**) levels were analyzed by Circos. Data are expressed as the mean ± SEM, *n* = 10 for each group. Two-way ANOVA: (**A**,**B**). Unpaired *t*-test: (**E**). * *p* < 0.05, ** *p* < 0.01.

**Figure 4 nutrients-14-04583-f004:**
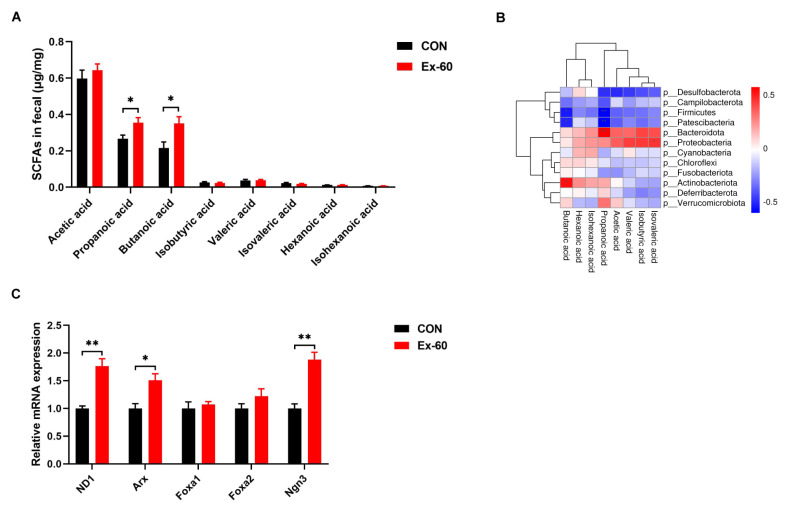
Two-week swimming training alters short-chain fatty acid profile. (**A**) SCFAs content in fecal samples, *n* = 8–10. (**B**) Pearson correlation analysis of SCFAs with gut microbiota on phylum level. Red indicates a positive correlation, blue indicates a negative correlation, and white indicates no correlation. (**C**) qPCR analysis of the expression of the regulators of L cell proliferation in colon samples. Data are expressed as the mean ± SEM. Unpaired *t*-test: (**A**,**C**). * *p* < 0.05, and ** *p* < 0.01.

**Figure 5 nutrients-14-04583-f005:**
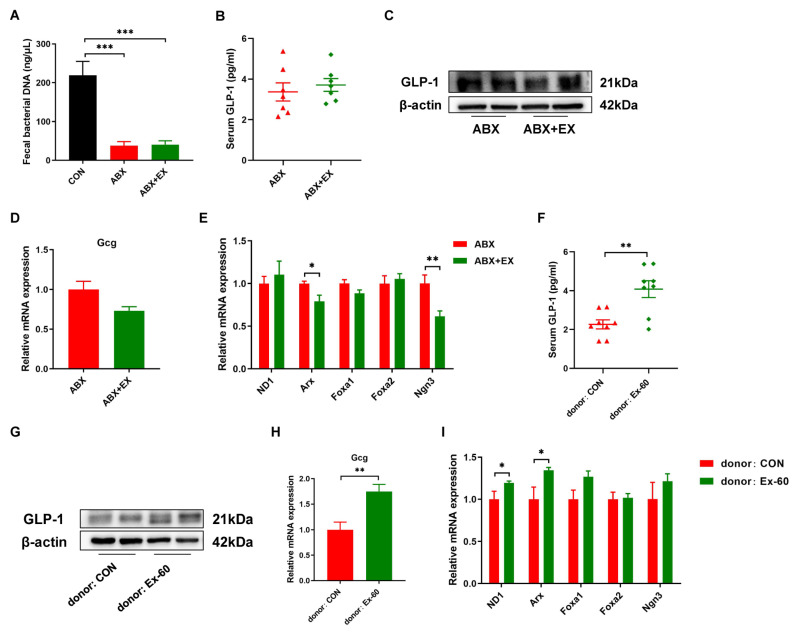
The gut microbiota contributes to the effects of exercise-induced GLP-1 secretion. (**A**) The concentration of fecal bacterial DNA. (**B**,**F**) GLP-1 levels were measured using an ELISA kit in serum. (**C**,**G**) Representative western blot of GLP-1 from colon samples. (**D**,**H**) qPCR analysis of the expression of the Gcg gene in the colon. (**E**,**I**) qPCR analysis of the expression of the regulators of L cell proliferation in the colon. Data are expressed as the mean ± SEM. One-way ANOVA: (**A**). Unpaired *t*-test: (**B**,**D**–**F**,**H**,**I**). * *p* < 0.05, ** *p* < 0.01, and *** *p* < 0.001.

**Figure 6 nutrients-14-04583-f006:**
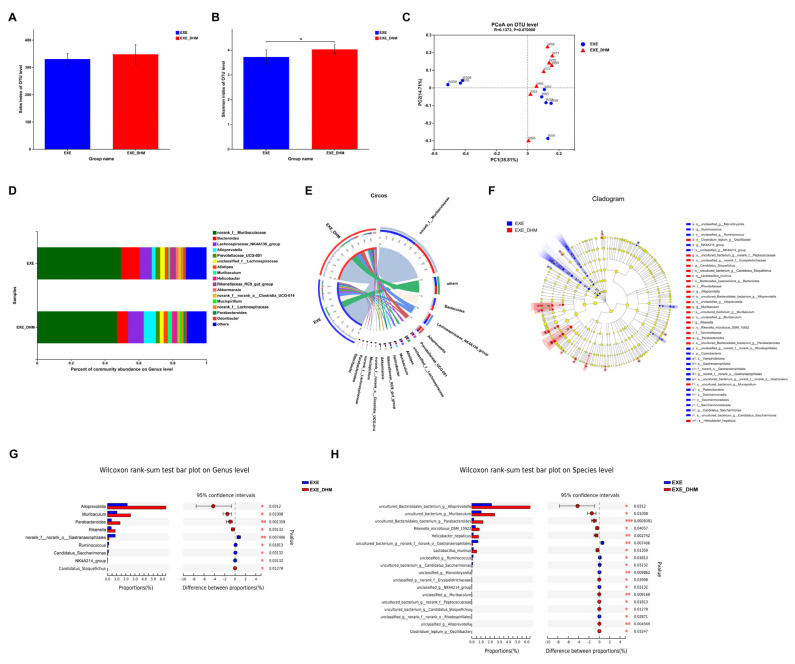
Dihydromyricetin alters the gut microbiota profile in exercised mice. (**A**,**B**) The OTU-level alpha diversity indices such as Sobs (**A**) and Shannon (**B**) index. (**C**) Weighted UniFrac PCoA analysis on OUT level. (**D**) Community composition analysis on genus level. (**E**) The relationships between samples and bacteria at the genus level were analyzed by Circos. (**F**) Linear discriminant analysis effect size (LEfSe) analysis from phylum to genus level. (**G**,**H**) Relative abundance and Wilcoxon rank-sum test bar plot on Genus (**G**) and Species (**H**) level. Data are expressed as the mean ± SEM. Unpaired *t*-test: (**A**,**B**). * *p* < 0.05.

**Figure 7 nutrients-14-04583-f007:**
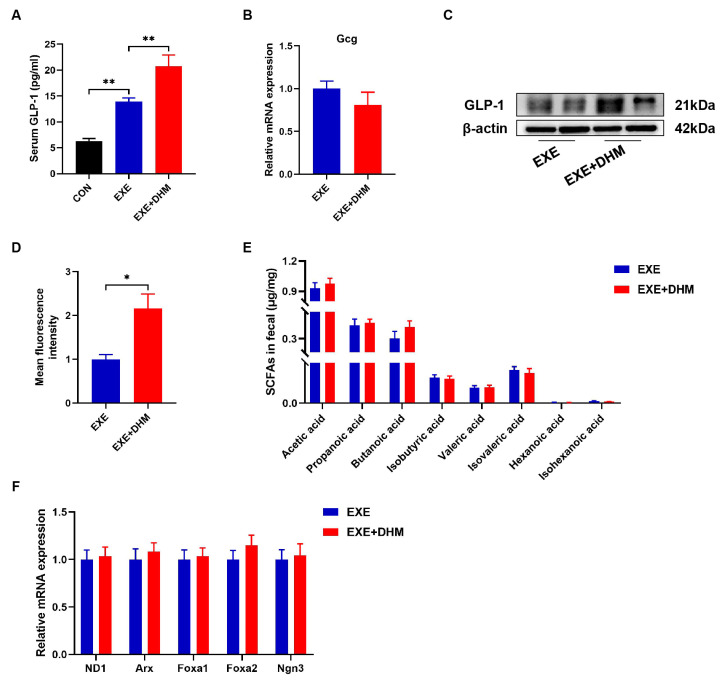
Dihydromyricetin enhances GLP-1 levels when combined with exercise. (**A**) GLP-1 levels were measured using an ELISA kit in serum. (**B**) qPCR analysis of the expression of the Gcg gene in the colon. (**C**) Representative western blot of GLP-1 from colon samples. (**D**) Analysis of GLP-1 positive expression in immunofluorescence staining of GLP-1 for colon samples using Image J. (**E**) SCFAs content in fecal samples, *n* = 8. (**F**) qPCR analysis of the expression of the regulators of L cell proliferation in the colon. Data are expressed as the mean ± SEM. One-way ANOVA: (**A**). Unpaired *t*-test: (**B**,**D**,**E**). * *p* < 0.05, and ** *p* < 0.01.

**Figure 8 nutrients-14-04583-f008:**
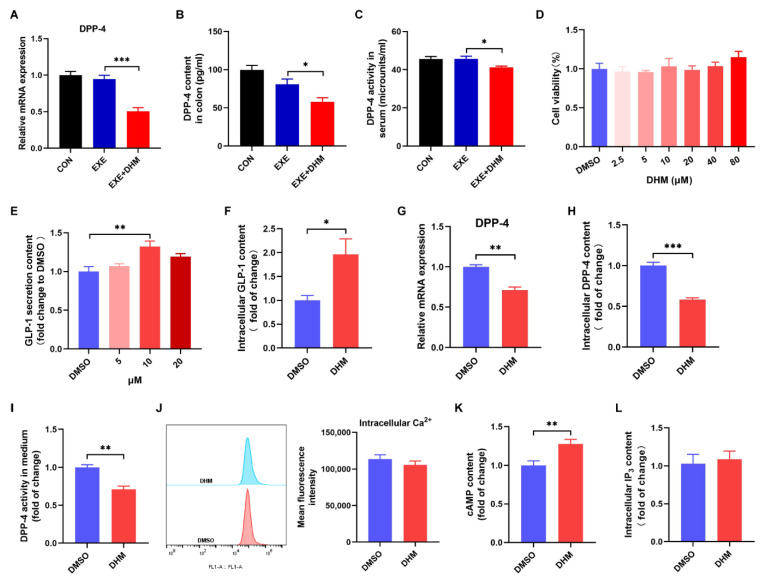
Dihydromyricetin stimulates GLP-1 secretion and inhibits DPP-4 in vivo and in vitro. (**A**) qPCR analysis of the expression of DPP-4 in the colon. (**B**) DPP-4 content was measured using an ELISA kit in the colon. (**C**) DPP-4 activity in serum. (**D**) Cell viability of GLUTag cells treated with DHM (0–80 μM). (**E**) GLP-1 secretion from GLUTag cells with DHM (5 μM, 10 μM, 20 μM) stimulation. (**F**) GLP-1 content in cell lysates was measured using an ELISA kit. (**G**) qPCR analysis of the gene expression of DPP-4 in GLUTag cells. (**H**) DPP-4 content in cell lysates was measured using an ELISA kit. (**I**) DPP-4 activity in cell culture medium. (**J**) Mean fluorescence intensity of intracellular Ca^2+^. (**K**) Luminescence indicated cAMP concentrations. (**L**) IP_3_ content in cell lysates was measured using an ELISA kit. Data are expressed as the mean ± SEM. One-way ANOVA: (**A**–**E**). Unpaired *t*-test: (**F**–**L**). * *p* < 0.05, ** *p* < 0.01, and *** *p* < 0.001.

**Table 1 nutrients-14-04583-t001:** Primers used for Real-time PCR.

Gene	Primer Sequence	
*Gcg*	Forward: TTACTTTGTGGCTGGATTGCTT	Reverse: AGTGGCGTTTGTCTTCATTCA
*DPP-4*	Forward: CCAATTCCAGAAGACAACCTTG	Reverse: CATCTGCCGTTCCATGAATAAG
*ND1*	Forward: GACGGGGTCCCAAAAAGAAAA	Reverse: GCCAAGCGCAGTGTCTCTATT
*Arx*	Forward: GGCCGGAGTGCAAGAGTAAAT	Reverse: TGCATGGCTTTTTCCTGGTCA
*Foxa1*	Forward: ACATTCAAGCGCAGCTACCC	Reverse: TGCTGGTTCTGGCGGTAATAG
*Foxa2*	Forward: CATGGGACCTCACCTGAGTC	Reverse: CATCGAGTTCATGTTGGCGTA
*Ngn3*	Forward: GCATGCACAACCTCAACTC	Reverse: TTTGTAAGTTTGGCGTCATC
*Actb*	Forward: GGCTGTATTCCCCTCCATCG	Reverse: CCAGTTGGTAACAATGCCATGT

Gcg, proglucagon; DPP-4, dipeptidyl peptidase IV; ND1, NeuroD1; Ngn3, neurogenin 3; Actb, β-Actin.

## Data Availability

The datasets analyzed during the current study are included in the article/Appendix A. Further inquiries can be directed to the corresponding author.

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
