# Peer review of "Dihydromyricetin Enhances Exercise-Induced GLP-1 Elevation through Stimulating cAMP and Inhibiting DPP-4"

_nutrients, 2022, doi:10.3390/nu14214583_

Round 1

Reviewer 1 Report

The study aims to examine whether endogenous hormone glucagon-like peptide-1 (GLP-1) could respond to exercise training in mice and the possible participation of microbiota, as well as whether dihydromyricetin (DHM) supplementation could enhance GLP-1 levels in response to exercise training. All these purposes are interesting to know the participation of acute aerobic exercise in the regulation of appetite, gastric emptying and energy intake. The experimental design involves animal ‘in vivo’ studies studying the effects of training, gut microbiota and DHN, but also involves murine enteroendocrine cell ‘in vitro’ cultures to investigate how DHM affected GLP-1 secretion and degradation. This experimental design is quite complicated, but it could be useful to attain the objectives.

Abstract: Indicate the name of GLP1

Line 89: What FMT experiments means?

Line 93: What was the dose of cocktail antibiotics intake the mice? We measured the water drunk by the mice?

Line 109: What about the CON group in relation t the two training weeks?

Line 119: In which sample did you analyse GLP1 concentration?

Line 159. ‘The extraction solutions were detected using an 8890B-5977B GC/MSD (Agilent Technologies Inc. CA, USA’. Describe the method used to analyse SCFAs in faecal samples.

Line 171: indicates a brief summary of the methods and units used to express the results.

Line 172: in which samples the DPP-4 content were analysed?

Line 177: In which samples the second messenger concentration was analysed? Line 186: When the One-way Anova Analysis was used? It could be better to use a Two-way ANOVA test in some experimental design as is the ‘3. To investigate the effects of a combined intervention (DHM and exercise)’.

Line 214, Figure 2. What did indicate the fold of change? Are the b-actin area taken into account in these values?

Line 216, Figure 2. Indicate the statistical test used.

Line 225 ‘As shown in Figure 3A and B, EXE_w2 mice had higher Sobs and Shannon index than CON_w2 mice’ The figures are poorly defined, practically the texts and the statistical markers in the figures are unreadable; Nevertheless, in the figure 3 A we can see significant differences between EXE_w2 and CON_w0 but not between EXE_W2 and CON_W2. In the figure 3 B perhaps, it could see significant differences between EXE_W2 and CON_W2, but it is difficult to see by these pictures.

Line 254. Indicate in Materials & Methods the statistical analysis performed to correlate these parameters. ‘To further screen out the gut microbiota associated with propionic acid and butanoic acid levels, Pearson correlation analysis was performed’.

Lines 265. In which samples were the gene expression determined? Indicate in Material & Methods the ARNm extraction procedure and also the gene expression quantification.

Lines 308-315. Correct typographical mistakes.

Figure 6 is unreadable

Figure 7. What does Figure 7D show?

Lines 359-361: ‘To test our hypothesis, we first examined the content and activity of DPP-4.  DHM treatment significantly decreased the gene and protein expression of DPP-4 in colon samples (Figure 8A, B)’ but the CON results are not shown. What about the CON results?

Line 361 and Figure 8C: Indicates in the figure legend the definition of microunits to express the DPP-4 activity.

Line 362. Indicates in M& M the cell viability procedure.

Lines 375-377: ‘This section may be divided by subheadings. It should provide a concise and precise description of the experimental results, their interpretation, as well as the experimental conclusions that can be drawn’. What this paragraph means? I agree with this comment regarding the way of writing this paragraph. Could you accommodate this correction request?

Line 487: The sentence ‘The increased SCFAs level activates Ffar2-PLCβ1 signalling…’ is not demonstrated in the results.

Line 492: The conclusion that ‘DHM is an exercise mimetics’ is not derived from your results.

Line 493: What ‘exercise enhancer’ means?

Author Response

Reviewer #1:

Dear Reviewer, We greatly appreciate your thoughtful comments that helped improve the manuscript. We trust that all of your comments have been addressed accordingly in the revised manuscript. Revised portion are marked in red in the paper. In the following, we give a point-by-point reply to your comments:

  1. Comment: Abstract: Indicate the name of GLP1.

Response: GLP-1 is known as glucagon-like peptide-1, and we have made correction in line 13-14 according to the reviewer’s comments.

  1. Comment: Line 89: What FMT experiments means?

Response: FMT experiments means fecal microbiota transplantation experiments, we have previously explained in line 66-67.

  1. Comment: Line 93: What was the dose of cocktail antibiotics intake the mice? We measured the water drunk by the mice?

Response: Antibiotic cocktail was added to drinking water with a mixture of the four antibiotics-0.5 g/L vancomycin, 1 g/L ampicillin, 1 g/L metronidazole, and 1 g/L neomycin sulfate. Mice were housed in groups of four per cage in the experiment 2-a. Thereby, water intake was calculated by dividing the measured intake by the number of mice in each cage. Results are shown in the Figure 1 attached below.

  1. Comment: Line 109: What about the CON group in relation to the two training weeks?

Response: At the two training weeks of EXE and EXE+DHM groups, the CON group maintained the natural state without swimming training.

  1. Comment: Line 119: In which sample did you analyse GLP1 concentration?

Response: We analyzed GLP-1 concentration in serum, cell medium supernatants, and cell lysates. We have added relevant information in line 119.

  1. Comment: Line 159. ‘The extraction solutions were detected using an 8890B-5977B GC/MSD (Agilent Technologies Inc. CA, USA’. Describe the method used to analyse SCFAs in faecal samples.

Response: We appreciate very much the reviewer’s concern. The chromatographic separation was carried out on a HP-FFAP capillary column (30 m × 0.25 mm × 0.25 μm, Agilent J&W Scientific, Folsom, CA, USA). The GC conditions were as follows: carrier gas (helium, 99.999%, flow rate 1 mL/min), injection temperature 260 °C, injection volume 1 µL in split mode with split ratio 10:1. The column temperature program was: initial temperature (80°C), raised to 120 °C at a rate of 40 °C/min, increased to 230 °C at a rate of 10 °C/min, and held at this temperature for 3 min. The MS parameters were as follows: ion source, electron impact (EI) at 230 °C; quadrupole temperature at 150 °C; transfer line temperature at 230°C ionization energy, 70 eV; ion source temperature, 230°C; and selected ion monitoring (SIM) mode. We have added relevant information in line 165-173.

  1. Comment: Line 171: indicates a brief summary of the methods and units used to express the results.

Response: DPP-4 content was expressed as pg/mL. DPP4 activity was reported as pm/min/mL= microunit/mL, one unit of DPP4 was the amount of enzyme that will hydrolyze the DPP4 substrate to yield 1.0 μm of fluorescent product (7-amino-4-methyl coumarin) per minute at 37℃. We have added relevant information in line 194, and 196-198.

  1. Comment: Line 172: in which samples the DPP-4 content were analysed?

Response: We analyzed DPP-4 content in colon and cell lysates. We have added relevant information in line 193.

  1. Comment: Line 177: In which samples the second messenger concentration was analysed? Line 186: When the One-way Anova Analysis was used? It could be better to use a Two-way ANOVA test in some experimental design as is the ‘3. To investigate the effects of a combined intervention (DHM and exercise)’.

Response: We appreciate very much the reviewer’s concern. In experiments on GLUTag cells the second messenger concentration was analysed. We have added relevant information in line 202-205. The two-way ANOVA test were performed to compare CON, EXE, and EXE+DHM group, as shown in Figure 7A, and 8A-C. We have made additional notes in line 223.

  1. Comment: Line 214, Figure 2. What did indicate the fold of change? Are the b-actin area taken into account in these values?

Response: GLP-1 protein expression was normalized to β-actin protein expression levels. The levels of GLP-1 protein in the swimming training groups were expressed as fold of changes relative to the level in the CON group.

  1. Comment: Line 216, Figure 2. Indicate the statistical test used.

Response: We added statistical analysis methods according to the reviewer’s comments in figure legend line 253-254.

  1. Comment: Line 225 ‘As shown in Figure 3A and B, EXE_w2 mice had higher Sobs and Shannon index than CON_w2 mice’ The figures are poorly defined, practically the texts and the statistical markers in the figures are unreadable; Nevertheless, in the figure 3 A we can see significant differences between EXE_w2 and CON_w0 but not between EXE_W2 and CON_W2. In the figure 3 B perhaps, it could see significant differences between EXE_W2 and CON_W2, but it is difficult to see by these pictures.

Response: To address the issue of low quality of Figure 3, we have provided higher resolution images of figure 3. It can be clearly seen from figure 3A and 3B that EXE_w2 mice had higher Sobs and Shannon index than CON_w2 mice.

  1. Comment: Line 254. Indicate in Materials & Methods the statistical analysis performed to correlate these parameters. ‘To further screen out the gut microbiota associated with propionic acid and butanoic acid levels, Pearson correlation analysis was performed’.

Response: The correlation among SCFAs and microbial populations were conducted by Pearson correlation analysis using R software version 3.5.1 and RStudio (R Package, USA), and a Pearson correlation of 0.5 or greater and a p-value<0.05 was considered as a significant correlation. We added the Pearson correlation analysis methods in Materials & Methods, line 153-156.

  1. Comment: Lines 265. In which samples were the gene expression determined? Indicate in Material & Methods the RNA extraction procedure and also the gene expression quantification.

Response: We measured the expression of L cell proliferation markers in colon tissues. According to the reviewer’s comments, we added the content in line 306. We previously put the real-time PCR procedure in the supplementary material and according to the reviewer' comment, we have added it clearly in M & M, line 208-217.

  1. Comment: Lines 308-315. Correct typographical mistakes.

Response: Here are the names of various gut microbiota. After careful checking, we found no typographical mistakes in this section.

  1. Figure 6 is unreadable.

Response: We have provided higher resolution images of figure 6 according to the reviewer' comment.

  1. Comment: Figure 7. What does Figure 7D show?

Response: Figure 7D represents the immunofluorescence staining of GLP-1 in the colon sample of the EXE and EXE+DHM groups. The red fluorescence indicates the positive expression of GLP-1, and the blue fluorescence indicates the nucleus. Moreover, the mean fluorescence intensity was quantified using Image J. We have added the content in line 388-389.

  1. Comment: Lines 359-361: ‘To test our hypothesis, we first examined the content and activity of DPP-4. DHM treatment significantly decreased the gene and protein expression of DPP-4 in colon samples (Figure 8A, B)’ but the CON results are not shown. What about the CON results?

Response: We measured the expression of DPP4 mRNA and DPP-4 content in the colon of CON group. We now have added the CON data in new Figure 8A, B and line 401-403.

  1. Comment: Line 361 and Figure 8C: Indicates in the figure legend the definition of microunits to express the DPP-4 activity.

Response: DPP4 activity was reported as pm/min/mL= microunit/mL, one unit of DPP4 was the amount of enzyme that will hydrolyze the DPP4 substrate to yield 1.0 μm of fluorescent product (7-amino-4-methyl coumarin) per minute at 37℃. We have added relevant information in M& M, line 196-198.

  1. Comment: Line 362. Indicates in M& M the cell viability procedure.

Response: We previously put the cell viability procedure in the supplementary material, and according to the reviewer' comment, we have added the cell viability procedure clearly in M& M, line 185-190.

  1. Comment: Lines 375-377: ‘This section may be divided by subheadings. It should provide a concise and precise description of the experimental results, their interpretation, as well as the experimental conclusions that can be drawn’. What this paragraph means? I agree with this comment regarding the way of writing this paragraph. Could you accommodate this correction request?

Response: The sentence “This section may be divided by subheadings. It should provide a concise and precise description of the experimental results, their interpretation, as well as the experimental conclusions that can be drawn.” is a part of a text given in the manuscript template. We're sorry we forgot to delete it. Now, we have deleted it.

  1. Comment: Line 487: The sentence ‘The increased SCFAs level activates Ffar2-PLCβ1 signalling…’ is not demonstrated in the results.

Response: We appreciate very much the reviewer’s concern. We have deleted the sentence.

  1. Comment: Line 492: The conclusion that ‘DHM is an exercise mimetics’ is not derived from your results.

Response: We appreciate very much the reviewer’s concern. We have deleted the sentence.

  1. Comment: Line 493: What ‘exercise enhancer’ means?

Response: Exercise mimetics and enhancer are a proposed class of therapeutics that specifically mimic or enhance the therapeutic effects of exercise [1-2]. Some small nutrient compounds, such as polyphenols, have been reported to have potential for development as exercise enhancers [3].

[1] Gubert C, Hannan AJ. Exercise mimetics: harnessing the therapeutic effects of physical activity. Nat Rev Drug Discov. 2021 Nov; 20(11): 862-879. doi: 10.1038/s41573-021-00217-1.

[2] Narkar VA, Downes M, Yu RT, Embler E, Wang YX, Banayo E, Mihaylova MM, Nelson MC, Zou Y, Juguilon H, Kang H, Shaw RJ, Evans RM. AMPK and PPARdelta agonists are exercise mimetics. Cell. 2008 Aug 8; 134(3): 405-15. doi: 10.1016/j.cell.2008.06.051.

[3] Craig DM, Ashcroft SP, Belew MY, Stocks B, Currell K, Baar K, Philp A. Utilizing small nutrient compounds as enhancers of exercise-induced mitochondrial biogenesis. Front Physiol. 2015 Oct 27; 6: 296. doi: 10.3389/fphys.2015.00296.

Reviewer 2 Report

This manuscript describes the investigation of the production of endogenous GLP-1 in response to exercise training in mice and  whether dihydromyricetin (DHM) supplementation can enhance GLP-1 levels in response to exercise training. Finally, a possible mechanism behind the GLP-1 secretion in response to exercise training is investigated. The manuscript contains some interesting and novel results that clearly is of interest for the readers of the journal.

The manuscript is well-structured and well-written. It gives a very nice introduction to the topic and provide with some interesting hypotheses that are investigated. The methods used in the present study, including in vivo studies, are sufficient described for reproducing the results, except for the GC-MS analysis of SCFAs (see comments below). Furthermore, the results are nicely presented both in text and figures and the discussion of the results are discussed in sufficient detail.

The results of this study demonstrates that exercise increases GLP-1 secretion, which is associated with an increase in the concentration of propanoic acid and butanoic acid due to an increase in the abundance of SCFAs-producing gut microbiota in mice. This is very nicely demonstrated in the manuscript and is perhaps not so surprising but on the other hand the results, which shows that increased SCFAs levels activates Ffar2-PLCβ1 signaling, which in turn stimulates L cells to secrete GLP-1 is interesting. Additionally, it is demonstrated that the naturally occurring flavonoid DHM interacts synergistically with exercise intervention to enhance GLP-1 levels through stimulating cAMP and inhibition of DPP-4, which is also interesting and may provide a deeper understanding of the pharmacological efficacies of DHM that can be used for the development of DHM as an exercise mimetics and exercise enhancer as also under stressed in the conclusions.

Overall, a well structured and well-written manuscript that contains some novel and interesting results. However, I have a few minor corrections and comments to the manuscript at described below:

Comments:

Page 4, line 153-161:

In the part of the Materials and methods section describing the measurement of SCFAs by GC-MS some information is missing about the column and the temperature program used for separating SCFAs. Please provide more details about the GC-MS analysis.

Page 12, line 375-377:

The sentence “This section may be divided by subheadings. It should provide a concise and precise description of the experimental results, their interpretation, as well as the experimental conclusions that can be drawn.” does not make sense and is probably a part of a text given in the manuscript template. Thus this sentence should be deleted.

Page 14, line 430-431:

It is not clear from the manuscript whether propanoic acid and butanoic acid are the only SCFAs that was detected or whether they were the only SCFAs that showed an increase in concentration in response to exercise and alterations in gut microbiota? If other SCFAs were detected/identified such as ethanoic acid, pentanoic acid, valeric acid, 3-methylbutanoic acid etc. they should also be mentioned in the manuscript even though their concentrations did not increase significantly in response to exercise.

Author Response

Responds to the reviewer’s comments:

Reviewer #2:

Dear Reviewer, We greatly appreciate your thoughtful comments that helped improve the manuscript. We trust that all of your comments have been addressed accordingly in the revised manuscript. Revised portion are marked in red in the paper. In the following, we give a point-by-point reply to your comments:

  1. Comment: Page 4, line 153-161:

In the part of the Materials and methods section describing the measurement of SCFAs by GC-MS some information is missing about the column and the temperature program used for separating SCFAs. Please provide more details about the GC-MS analysis.

Response: Thank you for pointing this out. The chromatographic separation was carried out on a HP-FFAP capillary column (30 m × 0.25 mm × 0.25 μm, Agilent J&W Scientific, Folsom, CA, USA). The GC conditions were as follows: carrier gas (helium, 99.999%, flow rate 1 mL/min), injection temperature 260 °C, injection volume 1 µL in split mode with split ratio 10:1. The column temperature program was: initial temperature (80°C), raised to 120 °C at a rate of 40 °C/min, increased to 230 °C at a rate of 10 °C/min, and held at this temperature for 3 min. The MS parameters were as follows: ion source, electron impact (EI) at 230 °C; quadrupole temperature at 150 °C; transfer line temperature at 230°C ionization energy, 70 eV; ion source temperature, 230°C; and selected ion monitoring (SIM) mode. We have added relevant information in line 165-173.

  1. Comment: Page 12, line 375-377:

The sentence “This section may be divided by subheadings. It should provide a concise and precise description of the experimental results, their interpretation, as well as the experimental conclusions that can be drawn.” does not make sense and is probably a part of a text given in the manuscript template. Thus this sentence should be deleted.

Response: The sentence “This section may be divided by subheadings. It should provide a concise and precise description of the experimental results, their interpretation, as well as the experimental conclusions that can be drawn.” is a part of a text given in the manuscript template. We're sorry we forgot to delete it. Now, we have deleted it.

  1. Comment: Page 14, line 430-431:

It is not clear from the manuscript whether propanoic acid and butanoic acid are the only SCFAs that was detected or whether they were the only SCFAs that showed an increase in concentration in response to exercise and alterations in gut microbiota? If other SCFAs were detected/identified such as ethanoic acid, pentanoic acid, valeric acid, 3-methylbutanoic acid etc. they should also be mentioned in the manuscript even though their concentrations did not increase significantly in response to exercise.

Response: We measured a total of eight SCFAs (acetic acid, propanoic acid, butanoic acid, isobutyric acid, valeric acid, isovaleric acid, hexanoic acid, isohexanoic acid) in the stools of CON and Ex-60 groups using GC-MS method. As shown in Figure 4A, Swimming training notably increased propanoic acid and butanoic acid levels, while the levels of other SCFAs displayed no differences between the CON and Ex-60 groups. We have added relevant information in line 289-292.

We tried our best to improve the manuscript and made some modifications in the manuscript. These modifications will not influence the content and frame work of the paper.

We appreciate for Reviewers’ warm work earnestly, and hope that the corrections will meet with approval.

Once again, thank you very much for your comments and suggestions.

Best regards!

Yours Sincerely,

Qianyong Zhang

2022-10-13

Round 2

Reviewer 1 Report

The manuscript has been satisfactory improved; however, some questions remained to be resolved.

Lines 260-264: You indicate to perform a two-way ANOVA analysis (Fig 3A,B) but you did not indicate the results of this statistical analysis. Was there a significant effect of time or was there a significant effect of exercise or was there an interaction between time and exercise? It might be better to specify these statistical results also in the figure, if it is possible to do so.

Lines 317-318: You indicate to perform a two-way ANOVA analysis on data on fecal bacterial DNA concentration (Figure 5A), but you did not indicate the results of this statistical analysis. Was there a significant effect of time or was there a significant effect of exercise or was there an interaction between time and exercise? It might be better to specify these statistical results also in the figure, if it is possible to do so.

Lines 376-377: You indicate to perform a two-way ANOVA analysis on data about GLP-1 levels (Figure 7A), but you did not indicate the results of this statistical analysis. Does this experimental design really support a two-way ANOVA?

Figure 7D: Nothing can be seen in this figure 7D. The dark background makes it difficult to see the blue or red fluorescence clearly. If it is not possible to improve it, it is better to eliminate it, leaving the bar diagram with the fluorescence intensities.

Line 428: Could you indicate the results of the two-way ANOVA analysis. Does this experimental design really support a two-way ANOVA? The effect of the time factor cannot be analyzed.

Line 533: the concept of ‘Exercise enhancer’ is very ambiguous and generally, it can be used for anything in relation to exercise. As indicated in his answer, it is a concept that derives from a previous publication (Front Physiol. 2015 Oct 27; 6: 296. doi: 10.3389/fphys. 2015.00296), but it specifies that it enhances exercise-induced mitochondrial biogenesis. Could you concrete what DHM enhance in relation to exercise? Or if it's not possible, it is better, eliminating this sentence.

Author Response

Dear Reviewer,

We greatly appreciate your thoughtful comments that helped improve the manuscript. We trust that all of your comments have been addressed accordingly in the revised manuscript. Revised portion are marked in red in the paper. In the following, we give a point-by-point reply to your comments:

  1. Comment: Lines 260-264: You indicate to perform a two-way ANOVA analysis (Fig 3A,B) but you did not indicate the results of this statistical analysis. Was there a significant effect of time or was there a significant effect of exercise or was there an interaction between time and exercise? It might be better to specify these statistical results also in the figure, if it is possible to do so.

Response: Thanks to the reviewer for pointing out this issue. The two-way ANOVA with independent variables being time and exercise is used for this study. We used a two-way ANOVA to analyze the main effect and interaction of time and exercise. As shown in Figure S1, there is no interaction between time and exercise (P>0.05). Since the focus of the experiment was to study whether exercise would change the composition of gut microbiota, we thought it might be better not to show the statistical results of two-way ANOVA analysis in Fig 3A and 3B. Moreover, for multiple group comparisons, we used the sidak's multiple comparisons test for individual effect analysis after two-way ANOVA analysis and showed the results in Fig 3A and 3B.

Figure S1 Results of two-way ANOVA

  1. Comment: Lines 317-318: You indicate to perform a two-way ANOVA analysis on data on fecal bacterial DNA concentration (Figure 5A), but you did not indicate the results of this statistical analysis. Was there a significant effect of time or was there a significant effect of exercise or was there an interaction between time and exercise? It might be better to specify these statistical results also in the figure, if it is possible to do so.

Response: Thanks to the reviewer for pointing out this issue. We have re-evaluated the statistical methods for the study design. Two-way ANOVA requires two independent variables, both of which are categorical variables, while one-way ANOVA was used to compare the differences between three or more groups of different treatment factors. Hence, one-way ANOVA is more appropriate in this experiment, owing to we regard ABX and ABX+EX as two different treatment ways. Moreover, the individual analysis of ABX+EX and ABX is what we want in our study. Therefore, we used tukey's multiple comparisons test for individual effect analysis between two groups after one-way ANOVA analysis. We have made correction in line 338-339.

  1. Comment: Lines 376-377: You indicate to perform a two-way ANOVA analysis on data about GLP-1 levels (Figure 7A), but you did not indicate the results of this statistical analysis. Does this experimental design really support a two-way ANOVA?

Response: Thanks to the reviewer for pointing out this issue. We have re-evaluated the statistical methods for the study design. Two-way ANOVA requires two independent variables, both of which are categorical variables, while one-way ANOVA was used to compare the differences between three or more groups of different treatment factors. Hence, one-way ANOVA is more appropriate in this experiment, owing to we regard EXE and EXE+DHM as two different treatment ways. Moreover, the individual analysis of EXE+DHM and EXE is what we want in our study. Therefore, we used tukey's multiple comparisons test for individual effect analysis between two groups after one-way ANOVA analysis. We have made correction in line 394.

  1. Comment: Figure 7D: Nothing can be seen in this figure 7D. The dark background makes it difficult to see the blue or red fluorescence clearly. If it is not possible to improve it, it is better to eliminate it, leaving the bar diagram with the fluorescence intensities.

Response: Thanks to the reviewer for pointing out this issue. The red and blue fluorescence can be clearly seen when the image is enlarged. We fully agree with Reviewer's opinion the dark background makes it difficult to see the blue or red fluorescence clearly in normal size. Considering the clarity of subsequent print, we eliminated the images of immunofluorescence staining, and left the bar diagram with the fluorescence intensities.

  1. Comment: Line 428: Could you indicate the results of the two-way ANOVA analysis. Does this experimental design really support a two-way ANOVA? The effect of the time factor cannot be analyzed.

Response: Thanks to the reviewer for pointing out this issue. We have re-evaluated the statistical methods for the study design. Two-way ANOVA requires two independent variables, both of which are categorical variables, while one-way ANOVA was used to compare the differences between three or more groups of different treatment factors. Hence, one-way ANOVA is more appropriate in this experiment, owing to we regard EXE and EXE+DHM as two different treatment ways. Moreover, the individual analysis of EXE+DHM and EXE is what we want in our study. Therefore, we used tukey's multiple comparisons test for individual effect analysis between two groups after one-way ANOVA analysis. We have made correction in line 430-431.

  1. Comment: Line 533: the concept of ‘Exercise enhancer’ is very ambiguous and generally, it can be used for anything in relation to exercise. As indicated in his answer, it is a concept that derives from a previous publication (Front Physiol. 2015 Oct 27; 6: 296. doi: 10.3389/fphys. 2015.00296), but it specifies that it enhances exercise-induced mitochondrial biogenesis. Could you concrete what DHM enhance in relation to exercise? Or if it's not possible, it is better, eliminating this sentence.

Response: Thanks to the reviewer for pointing out this issue. In experiment one, we found exercise significantly increased GLP-1 levels in mice. Additionally, in experiment three, our results indicated that DHM combined with exercise exhibited higher GLP-1 levels in mice, compared with the exercise alone. Combining the results of the experiment one and three, DHM enhances exercise-induced GLP-1 elevation, as the title suggests.

We tried our best to improve the manuscript and made some modifications in the manuscript. These modifications will not influence the content and frame work of the paper.

We appreciate for Reviewer’ warm work earnestly, and hope that the corrections will meet with approval.

Once again, thank you very much for your comments and suggestions.

Best regards!

Yours Sincerely,

Qianyong Zhang

2022-10-18
